# Corporate Governance Effects on Bank Profits in Gulf Cooperation Council Countries during the Pandemic

Hani El-Chaarani [1], Rebecca Abraham [2,*], Danielle Khalife [3] and Madonna Salameh-Ayanian [3]

1   Faculty of Business Administration, Tripoli Campus, Beirut Arab University, Riad El Solh,
    Beirut P.O. Box 1150-20, Lebanon
2   H. Wayne Huizenga College of Business and Entrepreneurship (HCBE), Nova Southeastern University,
    3301 College Ave, Fort Lauderdale, FL 33314, USA
3   Business School, Holy Spirit University of Kaslik, Jounieh P.O. Box 446, Lebanon
*   Correspondence: abraham@nova.edu

**Abstract:** During the COVID-19 lockdown, the typical bank in the Middle East lost liquidity due to deposit drains and experienced increases in nonperforming loans. The loss of liquidity was a supply shock, while the increase in nonperforming loans was a demand shock. Corporate governance increases the board's oversight of top management's implementation of strategies to reduce these shocks. Two corporate governance measures include a political concentration in the ownership and the presence of independent directors on the board of directors. Politically connected shareholders can ensure the continuous flow of deposits through their access to large depositors, thereby reducing supply shocks. Supply shocks may also be overcome by the large deposit balances from oil wealth. Independent directors are not employees of the banks on whose boards they serve, thereby providing objective evaluations of management's performance. Managers who are evaluated by independent directors can reduce nonperforming loans by strictly evaluating the creditworthiness of borrowers and providing incentives for timely repayment. Thus, demand shocks may be overcome by the scrutiny of management by independent directors. These conditions prevail in the Gulf Cooperation Council (GCC countries). Using a sample of 326 GCC banks, we perform OLS regressions followed by two-stage least squares and GMM estimator robustness checks of ownership's political concentration, independent directors, bank size, and bank liquidity on returns on assets and equity. Ownership political concentration, independent directors, bank size, and liquidity ratio significantly explained the return on assets and on equity. We conclude that large shareholders use political connections to cope with crises and that large banks are able to make new loans due to liquidity from large reserves. Independent directors evaluate management performance objectively, thereby requiring that management reduce nonperforming loans. We close research gaps of bank performance in GCC countries, as opposed to the entire MENA region, the latter being the focus of the literature. The significance of this paper is that it demonstrates the ability of banks to employ corporate governance to cope with crises. This is an original approach, as it seeks the outcome of a positive signal on bank performance of the reduction in the supply shock through ownership political concentration and reduction in the demand shock by independent directors. As corporate governance variables mitigate both shocks, corporate governance may assist banks in coping with liquidity crises.

**Keywords:** corporate governance; GCC region; banking; COVID-19 crisis; bank performance

## 1. Introduction

As financial intermediaries, banks accept deposits, which supply the cash needed to fund loans. The smooth operation of both supply of funds, and demand for funds, is fundamental to intermediation, as there must be sufficient deposits to create loans and creditworthy borrowers for these loans. Such was the normal conduct of banking, both in developed countries and developing countries. In the MENA (Middle East and North

Africa) region, banks financed trade by making loans to exporters. The deposit base (particularly in the Gulf Cooperation Council (GCC) countries, which are a subset of the MENA region) was robust, with businesses making regular deposits and oil exporters making large deposits.

The outbreak of the highly infectious disease, COVID-19, in 2020 ushered in a black swan event. To protect against transmission of the disease, schools, businesses, and retail stores closed, resulting in the cessation of face-to-face business activity, widespread unemployment, and stay-at-home work. Aggregate demand, production, and foreign trade declined sharply (Ghosh and Saima 2021). As financial intermediaries that convert deposits to loans, banks experienced both supply shocks and demand shocks. Supply shocks originated from deposit drains, as businesses and individuals withdrew savings to meet short-term liquidity needs. Businesses lost revenue from site closures and from revenue deferrals by business customers who had lost customers due to closures. Businesses needed bank deposits to pay fixed expenses, such as rent and utilities, that were paid from regular revenue in the pre-pandemic era. Individuals withdrew savings to supplement incomes diminished by unemployment. Such deposit drains reduced the number of loans that banks could issue. Prior to COVID-19, the MENA region was characterized by large volumes of nonperforming loans (Mdaghri 2022). Mdaghri (2022) documented the loss of confidence by depositors in banks with high levels of nonperforming loans. This trend was exacerbated by the pandemic. Business borrowers, unable to collect trade credit from customers, defaulted, as did individuals who lost income from unemployment. Banks experienced losses in interest income and losses in loan principal from loan defaults. Demand for loans from creditworthy borrowers declined sharply, ushering in a demand shock. These outcomes occurred across the Middle East and North Africa (MENA) region (El-Chaarani et al. 2022).

However, banks in the GCC (Gulf Cooperation Council) component of the MENA region may have coped with the COVID-19 crisis more effectively. The GCC consists of Bahrain, Kuwait, Qatar, Oman, United Arab Emirates, and Saudi Arabia. It is unique in that it consists of oil exporters with large cash reserves. Intuitively, a proportion of these cash reserves form the core deposits of large banks. Further, oil exporters may supply equity capital to banks, thereby becoming large shareholders. Therefore, the liquidity needed for bank deposits could have been maintained during the COVID-19 lockdown, thereby partly mitigating the supply shock to bank liquidity. However, the demand shock to banks of borrower defaults may have been unchanged.

Bank stability has been associated with good corporate governance. Corporate governance is practiced by board members who view themselves as stewards of the firm (Donaldson and Davis 1991). It consists of board action to enhance the firm's wealth. Measures include (1) a large board, which encourages a diversity of viewpoints, (2) independent directors on the board, who objectively evaluate management, (3) ownership by large shareholders, institutions, and foreigners, who reduce agency conflicts, (4) compensation-based performance, (5) ownership's political concentration (large shareholders with relationships with politicians), and (6) managerial political connections. Fu et al. (2014) observed that effective corporate governance increases bank profits, reduces risk, creates value, and promotes efficient operations. It follows that some or all of these corporate governance variables may reduce the harmful effects of supply shocks or demand shocks on bank profitability during a financial crisis, such as the COVID-19 lockdown.

Institutional variables that influence bank performance include bank size and liquidity. Large shareholders gravitate to large banks. Given that large banks are symbols of economic stability, they are likely to receive government bailouts during crises, ensuring their access to loanable funds. Large banks also have access to business leaders, international organizations, and foreign depositors, all of which may provide deposits. The large volume of such deposits increases the total number of loans that may be created or increases bank liquidity.

In this paper, we explore the effects of certain corporate governance variables, and institutional variables on bank profits, measured as return on assets and return on equity.

We expect that these variables will reduce the adverse effects on banks of supply shocks and demand shocks. They include the corporate governance variables of the ownership's political concentration, managerial political connections, and independent directors. Large shareholders may have political connections with powerful individuals in government, such as ministers, who provide access to government sources of funding, and loan creation from large-scale infrastructure projects. Thus, large shareholders may maintain liquidity, even during crises, reducing the negative effects of supply shocks. The coalition hypothesis views large shareholders and politicians as forming a coalition that safeguards the interest of banks (Boussada and Hakimi 2021). The presence of independent directors on the boards of directors of banks may result in objective evaluations of top management's ability to reduce nonperforming loans. Such a reduction of defaults partly mitigates the demand shock of a financial crisis. Managerial political connections, or managerial relationships, may adversely affect bank profits, as managers may hire underperforming friends of politicians or invest in low-NPV projects supported by politicians. In order to examine both types of shock on bank performance, we measured the effects of the ownership's political concentration, bank size, liquidity creation, managerial political connections, and independent directors on banks in GCC countries during the 2020–2021 COVID-19 lockdown. We envision the ownership's political concentration, bank size, and liquidity creation as antidotes to supply shocks of fewer loans, while independent directors, and lack of managerial political connections, cope with demand shocks through efficient cost containment of issuing new financial products.

We advance knowledge in four key areas. The first area is banking in the MENA region. The literature on bank performance in the MENA region is fragmented, with studies on liquidity creation, bank size, and nonperforming loans. Key studies were performed by Mohammed (2014) and Boussada and Hakimi (2021). Mohammed (2014) examined liquidity creation in banks in 18 MENA countries, finding that large banks created more liquidity than small banks and SME banks. Liquidity creation had a positive impact on the return on assets for large banks and an adverse effect on the return on assets for their small and medium-sized counterparts. Boussada and Hakimi (2021) obtained mixed results, with support for the dispersion hypothesis, whereby multiple large shareholders reduced profitability, and for the coalition hypothesis, with a few large shareholders increasing profitability. Other studies, such as Mdaghri (2022) and Sahyouni and Wang (2019), underscored the reduction in profitability from nonperforming loans due to the loss of confidence on the part of depositors with high nonperforming loans. They cautioned against excessive liquidity creation, which increased the percentage of nonperforming loans, with unchanged credit restrictions. To achieve coherence in the assessment of bank performance, our variables are lodged in a corporate governance framework. This framework employs board characteristics, such as independent directors, ownership concentration, and managerial governance characteristics, such as managerial political connections. Corporate governance is particularly important for banks during crises, as the financial crisis of 2007–2008 demonstrated that banks with strong governance structures provided protection for depositors (Khediri et al. 2021) and built public trust (El-Chaarani and Abraham 2022). Governance structures were particularly effective in curbing the extraction of private benefits from banks by politicians (Eichler and Sobański 2016; La Porta et al. 2002). We conjecture that the differential financial performance of different GCC banks may have been due to differential governance structures.

Although contributing to the literature on the MENA region, this paper views the GCC as a distinct entity within the MENA region. The MENA region consists of the GCC countries and economies that trade agricultural commodities with former colonial powers. The GCC has oil wealth, which bestows higher incomes upon its citizens. The rest of the MENA region has lower incomes from non-oil commodity trade. In other words, the oil wealth and higher incomes of GCC countries set them apart from the remainder of the MENA region. With undeveloped capital markets throughout the region, banks provide funding for private-sector growth and act as the conduit for the distribution of government

funding for the public sector. Yet, during a crisis, GCC banks are in a stronger position due to their large reserves, unlike the rest of the MENA region, which has no such source of cash deposits. During crises, while corporate governance in other MENA countries emphasizes the monitoring of bank management's ability to increase liquidity for bank loans, in the GCC, independent directors may require that management reduce nonperforming loans.

The second research gap lies in the influence of political connections on bank performance. The current knowledge of such connections is either anecdotal or confounded by a country (Lebanon) that was in crisis due to non-COVID-19 lockdowns. Attalah and Tamo (2021) enumerate the ownership of up to 54% of a Lebanese bank owned by two family members, while two former government ministers became the majority shareholders of a bank. A single study of Lebanese banks in 2021–2022 determined empirically that the presence of independent members on the board of directors, as well as the presence of audit, risk, and compliance committees, increased profitability. However, political connections increased the level of nonperforming loans (El-Chaarani and Abraham 2022). We expand both the sphere (the entire GCC) of examination and separate non-COVID-19 crises, such as the Lebanese financial crisis of 2021, from pandemic-induced crises.

From a practical standpoint, this study could assist bankers in identifying the ideal governance strategies that are consistent with the level of legal protection adopted. This study could identify the optimal governance structure that enhances performance and mitigate the negative impact of crises.

The remainder of this paper is organized as follows. Section 2 is a Review of the Literature; Section 3 consists of hypotheses development; Section 4 is Methods and Materials; Section 5 is Results; Section 6 consists of Conclusions.

## 2. Review of Literature

In order to contextualize the effect of corporate governance on banks during financial crises, we first review the literature on corporate governance in the pre-pandemic period and during the pandemic. Then, we explore the effects of corporate governance on bank financial crises in a broader context during the financial crisis of 2007–2009. We contend that this approach provides information on corporate governance in banks while highlighting the measures taken by banks to cope with an earlier financial crisis. We refer to papers on banking in the Middle East; however, the paucity of such work requires us to examine research from other locations.

### 2.1. The Influence of Corporate Governance on Banks

The Pre-COVID-19 Era: Ownership concentration is the presence of large shareholders on the boards of directors of banks. Intuitively, the presence of large shareholders on the boards of banks may be beneficial. If large shareholders influence other shareholders to uphold shareholder wealth maximization as a goal, the board will make rational decisions that promote the long-term interests of the firm. We review three studies in the pre-COVID-19 era that describe the impact of ownership concentration in the MENA region.

Using a sample of MENA banks in the 2001–2012 period, Haque (2019) found that ownership concentration increased bank risk-taking. Capital stringency further increased risk-taking, as scarce capital was employed in low-NPV projects. Risk-taking was curbed by activity restrictions. Activity restrictions restricted the freedom of large shareholders to reject unprofitable projects. Otero et al. (2020) used a slightly more contemporary sample from 2005–2012, reinforcing Haque's (2019) finding that ownership concentration increased bank risk-taking, as shareholders sought higher returns from risky projects, regardless of insolvency risk. Moral hazard prevailed with strong bank regulation and law enforcement in the country, freeing banks from responsibility for the consequences of excessive risk-taking. In other words, regulatory measures, such as deposit insurance, freed bank managers from cautiously using deposits, as they knew that government would indemnify depositors if their funds were lost through excessive risk-taking. In the most current of these studies, with data from the 2011–2018 period, Issa et al. (2021) observed

that board diversity by nationality significantly increased bank returns on assets and on equity. This was an early study of governance on bank profitability rather than risk-taking. In short, governance may increase bank risk-taking while having a positive influence on profitability.

The COVID-19 Lockdown Period: Likewise, opposing findings have emerged from examinations of corporate governance during the COVID-19 crisis. Broadstock et al. (2021) observed that internal committees, along with a well-structured board, encouraged banks to become innovative and profitable. Other boards communicated with external stakeholders, such as depositors, thereby increasing trust. In contrast, Demir and Danisman (2021) conducted a cross-national study in 2020 with 110 banks, which failed to observe relationships between corporate governance and bank returns. Takahashi and Yamada (2021) did not obtain any significant influence of corporate governance mechanisms on Japanese stock returns.

El-Chaarani et al. (2022) obtained mixed results, with certain corporate governance mechanisms significantly increasing profitability and other corporate governance measures yielding no effect on profitability. In an examination of MENA banks during the COVID-19 pandemic period, they observed that the presence of independent directors on the board of directors and high ownership concentration significantly influenced bank profitability. Performance-based compensation, the presence of women on boards, and anti-takeover provisions had no effect on profitability. Certain institutional variables, such as lack of political pressure on board members and strong legal protection, increased bank profitability.

In sum, in the MENA region, COVID-19 restrictions increased the positive influence of certain corporate governance mechanisms, such as ownership concentration and independent directors on the board. Institutional variables, such as bank regulation and law enforcement, decreased profitability, while lack of political pressure on board members and legal protection increased bank profitability.

*2.2. The Effect of Corporate Governance Mechanisms on Banks during the 2007–2009 Financial Crisis*

The 2007–2009 financial crisis was partly attributed to the collapse of the control mechanisms that prevented excessive bank risk-taking. As corporate governance mechanisms impose restrictions on excessive risk-taking, their presence during the crisis was expected to reduce losses, while their presence after the crisis was expected to speed recovery. Results from three studies have been mixed, with certain corporate governance measures positively influencing bank returns while others failed to improve bank performance.

In an examination of Spanish banks during the crisis, Ayadi et al. (2019) observed that (1) independent directors were associated with negative returns on assets, (2) compensation committees increased return on assets, (3) CEO duality (CEO as board chair) increased return on assets. Independent directors' ability to objectively evaluate managers is an important characteristic of good corporate governance, so the above result is puzzling. The authors reexamined the analysis, finding that having a majority of independent directors increased return on assets and encouraged other directors to be impartial in evaluating management's ability to cope with the crisis. Compensation committees that support fair compensation for CEO attempting to recover from the crisis improved bank profits. During the crisis, CEOs serving as board chairs suppressed agency conflicts, instead using their power as board chairs to harness the creativity of top management and board members to develop novel solutions to emerge from the crisis.

Bachiller and Garcia-Lacalle's (2018) assessment of Spanish banks during the 2007–2009 financial crisis found that the politicization of board members had no effect on bank financial performance, board size increased social responsibility, and board support of social welfare programs increased profitability. Perhaps, political connections were not effective in obtaining preferential government funding. Board support of social welfare programs funded by the government was to be expected, with 50% of the banks being recipients of government bailouts.

In an assessment of Russian banks, Orazalin et al. (2016) observed governance characteristics, such as the number of directors, independent directors, and monitoring committees, had no effect on bank performance during the crisis. However, the existence of these corporate governance mechanisms assisted banks in emerging from the crisis, presumably because board oversight of management mandates that management adopts creative solutions to finding bank liquidity and limiting excessive risk-taking.

In summary, Table 1 shows the corporate governance variables that have been found to affect bank profits during both the 2007–2008 financial crisis and the COVID-19 lockdown. Five board mechanisms, i.e., internal committees, independent directors, ownership concentration, compensation committee, and CEO duality, were found to influence bank profits during financial crises.

**Table 1.** Summary of the Literature on Corporate Governance on Bank Performance During Crises.

| Corporate Governance Characteristic | 2007–2009 Financial Crisis | COVID-19 Lockdown |
|---|---|---|
| Internal Committees | | Broadstock et al. (2021) Increase bank profits |
| Independent Directors | Ayadi et al. (2019) | El-Chaarani et al. (2022) |
| | Decrease bank profits | Increase bank profits |
| Ownership Concentration | | El-Chaarani et al. (2022) Increase bank profits |
| Board Support of Social Welfare | Bachiller and Garcia-Lacalle (2018) Increase bank profits | |
| Compensation Committee | Ayadi et al. (2019) Increase bank profits | |
| CEO Duality | Ayadi et al. (2019) Increase bank profits | |

## 3. Hypotheses Development

### 3.1. Independent Directors and Bank Profitability

By definition, independent directors cannot be employees of the banks on whose boards they are members. They cannot have material interests in the bank. This suggests that they are likely to be impartial in their assessment of bank management, as they do not personally know members of the senior management team. Such unbiased assessments are crucial during a pandemic when independent directors must have the freedom to evaluate the strategies being employed to seek alternative revenue streams and reduce the surge in nonperforming loans. Independent directors are in a position to curb managers who engage in ineffective new strategies while supporting revenue-enhancing measures.

What is the specific role of independent directors during a crisis? Independent directors may reduce supply shocks by monitoring management's ability to maintain liquidity levels for deposit creation. As an example, Arora (2018) evaluated bankrupted banks, observing that independent directors outperformed their peers in searching for information, giving advice, and accessing needed capital. In addition, independent directors must monitor top management's ability to reduce increases in nonperforming loans. Reddy and Locke (2014) observed that large shareholders, in conjunction with independent directors, demanded that management employ capital for shareholder wealth maximization rather than their own preferred projects. Independent directors have been found to prevail upon management to increase the quality of risk disclosure (Zulfikar et al. 2017), and disclosure of CSR (Rouf and Hossan 2021). In banks, demands for disclosure take the form of revealing the level of nonperforming loans. To avoid embarrassment from the revelation of high levels of nonperforming loans, management may embark on strategies to reduce the level of nonperforming loans. Both maintenance of liquidity levels and reduction of

nonperforming loans permit the continuation of bank operations or an increase in bank profits. Handriani and Robiyanto's (2019) examination of Indonesian firms supported the link between independent directors and profitability in that independent directors significantly increased Tobin's Q.

**Hypothesis 1.** *The presence of independent directors on the board of directors increased bank profitability among GCC banks before the COVID-19 crisis and during the COVID-19 crisis.*

*3.2. Ownership Concentration (Institutional, Foreign, and Political) and Bank Profitability*

Bank managers have responsibilities to multiple entities, including regulators, shareholders, and depositors. Managers must be evaluated in their ability to meet the needs of all of these parties rather than serving their own interests. Certain banks have large shareholders on the board. Boussada and Hakimi (2021) presented the coalition hypothesis, whereby large shareholders on bank boards form a coalition that monitors managerial performance. Such a coalition reduces agency costs by dissuading management from making unprofitable decisions. Senior management is likely to comply as their rewards and job security depend upon the support of the board. Constructive dissent among large shareholders may result in the most optimal course of action being chosen, i.e., by subjecting alternatives to rigorous evaluation, suboptimal choices may be eliminated. The benefits of improved control of CEOs by large shareholders, greater transparency (Sivaprasad and Matthew 2021), improved operational efficiency, and risk management (Unite and Sullivan 2003), have been documented.

Who are the large shareholders? Large shareholders may be politicians or close friends and family members of politicians. Government ministers and high-ranking military personnel may become shareholders of banks. If they acquire significant amounts of stock, they become large shareholders. As owners of the bank, they wish to increase the value of bank stock and, in turn, their own wealth. These politician-shareholders promote shareholder wealth maximization. Such boards may seek alternative sources of revenue during a crisis by hiring experts. El-Chaarani et al. (2022) examined board activity in the MENA region during the COVID-19 lockdown. They found that bank boards contained former members of the military, and government ministers, who used their political connections to steer funds to banks with which they were associated. Large shareholders helped to maintain liquidity levels, preventing the intensification of liquidity supply shocks. Both returns on assets and return on equity increased in these banks. This result was further supported by the significance of the ownership's political concentration in explaining bank return on assets and bank return on equity during the 2021 Lebanese financial crisis (El-Chaarani and Abraham 2022).

Other large shareholders include institutional investors and foreign investors. Large shareholders may represent financial institutions with large asset bases, such as pension funds and life insurance companies. Such shareholders may be effective in monitoring managers due to their access to vast resources and expertise in asset management and financial reporting. Foreign investors on bank boards may have a similar effect to institutional investors. As leading investors and business owners in their home countries, they are often skilled in project evaluation, bank regulation, and management of banks in crisis. In bank-oriented countries, where banks supply large businesses with capital, close relationships between government and large businesses bestow negotiation skills and regulatory knowledge upon business owners, who may become foreign investors on bank boards in other countries. Barry et al. (2011) found that the existence of institutional and foreign owners was associated with a decrease in bank asset risk and a decrease in bank default risk during the 2007–2008 financial crisis. Such risk reductions may result in greater bank profitability.

**Hypothesis 2.** *Ownership concentration on the board of directors increased bank profitability among GCC banks before the COVID-19 crisis and during the COVID-19 crisis. Specifically, owners may be institutional investors, foreign investors, or politically connected individuals.*

*3.3. Managerial Political Connections and Bank Profitability*

Managerial political connections exist at the middle management and operating management levels. These managers do not have the responsibility of providing visionary leadership to take the bank out of the financial crisis. Their interests are more immediate, with hiring and resource allocation being their leading concerns. Intuitively, managers who have close ties with politicians may favor the hiring of the friends and family of these politicians, regardless of their qualifications. Such hiring of unskilled individuals restricts the bank's ability to draw on the managerial talent required to cope with crises. La Porta et al. (2002) argued that the presence of politicians in management positions at banks decreased public trust in the legal protection offered to them in the event of disputes with banks, which in turn, adversely affected financial performance. El-Chaarani and Lombardi (2022) showed that politicians encouraged middle managers to engage in excessive lending to the Lebanese government. Middle managers and operating-level managers have been shown to demonstrate moral hazard (Braham et al. 2020). Moral hazards occur when bank loan officers make excessively risky loans, as they feel that deposit insurance will protect depositors. The reasoning is that depositors will not lose deposits in the event of loan defaults, as deposit insurance will pay the full amount of their deposits. As managers are no longer accountable for loan defaults, they will make risky loans. Moral hazard in lending was uncovered by Braham et al. (2020), who observed that banks with politically connected managers took excessive risks in lending, as they felt that the politicians would bail them out in the event of defaults on the loans. Risky loans increase the severity of the demand shock of nonperforming loans during a financial crisis. Such loan defaults would adversely affect bank profits.

**Hypothesis 3.** *Managerial political connections decreased bank profitability among GCC banks before the COVID-19 crisis and during the COVID-19 crisis.*

*3.4. Other Corporate Governance Variables and Bank Profitability*

The literature has shown that CEO Duality and Compensation-Based Performance significantly influence bank profitability. Therefore, we included them as predictors of bank profitability, both before the COVID-19 crisis and during the COVID-19 crisis.

CEO Duality is a bank CEO performing the dual role of chairman of the board. The two roles elevate CEOs to high levels of internal control. As CEOs, they have access to privileged internal information about bank performance. As board chairs, they are in a position of power over other directors. CEOs who are entrenched exploit the information asymmetry of having more knowledge of the bank's finances than shareholders. They may divert bank loans to low NPV projects that favor their own interests. The power of the chairman of the board prevents other board members from objectively monitoring managers, who are the CEO's associates. In Yu's (2022) review of 314 empirical studies of banks, CEO duality was an impediment to the objective evaluation of management.

Compensation-based performance can be used to align the interests of the CEO with those of the bank. Sun (2014) showed that rewarding CEOs based on performance increased the profitability of banks during the 2007–2008 crisis, as CEOs refrained from excessive risk-taking, which would reduce bank profitability. They found alternative revenue streams that boosted shareholder returns. Ayadi et al. (2019) obtained a similar result during the 2007–2009 financial crisis. Well-compensated CEOs, whose compensation was determined by a compensation committee, proposed measures to sustain bank profitability.

*3.5. Bank Size*

Large banks in the GCC region receive large deposits from oil companies and affluent individuals. These deposits could provide the source of cash needed to fund new loans. In addition, large banks accumulate cash reserves over time. They have numerous retail banking locations, which accept deposits from many businesses and individuals. These deposits may also be used to fund mortgages, auto loans, credit cards, and personal loans without interruption during a financial crisis. Large banks have preferential access to alternative revenue streams, such as forward contracts, futures contracts, letters of credit, and loan commitments. Businesses that employ these financial instruments may prefer to obtain them from large banks, as they are certain that the bank has the funds to complete these transactions. For example, if a business needs a letter of credit, which is a bank guarantee that the bank will pay a vendor in the event that the business does not have the funds, the business may seek the letter of credit from a large bank that has the funds to provide such a guarantee.

The literature offers the economies of scale argument with a large deposit base belonging to large banks (Athanasoglou et al. 2008; Flamini et al. 2009). The large deposit base helps banks to grow over time. Adelpo et al. (2018) observed that bank size was associated with profitability for a sample of West African banks before, during, and after the financial crisis of 2007–2009. Bank size was one of the few predictors of bank profitability that did not depend on the period of analysis or measure of bank profitability used. Gupta and Mahakud (2020) found that bank size predicted profitability for a sample of Indian banks during the 2007–2009 crisis. Like the Adelpo et al. (2018) study, this result remained robust with the measures used.

**Hypothesis 4.** *Bank size increased bank profitability among GCC banks before the COVID-19 crisis and during the COVID-19 crisis.*

*3.6. Liquidity Ratio*

Banks are dependent on the creation of liquidity. In other words, banks must be effective intermediaries, converting deposits to loans. This is the essential function of a bank, which may be severely impeded during a crisis when both deposits and loans decline, as during the COVID-19 lockdown. The GCC countries may have fared better than the typical MENA country bank, as their oil exports continued to generate bank deposits, even under deteriorating macroeconomic conditions. Essentially, liquidity maintains financial stability, even during the deteriorating market conditions of a financial crisis. Adelpo et al. (2018) recommend short-term lending during a financial crisis to ensure that loans are repaid in sufficiently large amounts in a short time, thereby ensuring that sufficient cash flows are received by banks.

**Hypothesis 5.** *Bank liquidity ratios increased profitability among GCC banks before the COVID-19 crisis and during the COVID-19 crisis.*

**4. Methods and Materials**

*4.1. Data and Sample Characteristics*

The sample of this proposal includes all of the commercial banks that exist in GCC from 2018 to 2021, namely in Bahrain, Kuwait, Oman, Qatar, United Arab Emirates, and Saudi Arabia. We believe that this period is appropriate as 2018 experienced some decline in bank revenues due to rumors of the forthcoming pandemic from travelers to Wuhan, China. The COVID-19 pandemic began in the last quarter of 2019; lockdowns persisted through 2020 and 2021. We excluded Islamic and other bank types due to their specific characteristics and their Islamic governance structure.

The financial data of the banking sector was derived from Orbis Bankscope Database. The corporate governance and political connection information were collected from annual bank reports. We excluded any bank that had incomplete financial information. GDP was

extracted from the World Bank database. In total, the number of banks is 164 from six countries (see Table 2). The largest number of banks were in Bahrain (31.90%) and United Arab Emirates (21.78%). The final dataset consisted of 326 bank/year observations from 6 countries.

**Table 2.** Description of the Sample.

| Country Name | Banks Observations 2018 | Banks Observations 2019 | Banks Observations 2020 | Banks Observations 2021 | Total Observation per Country | Percentage per Country |
|---|---|---|---|---|---|---|
| Saudi Arabia | 9 | 8 | 8 | 7 | 32 | 9.82% |
| Qatar | 8 | 6 | 7 | 6 | 27 | 8.28% |
| Oman | 17 | 15 | 13 | 16 | 61 | 18.71% |
| The United Arab Emirates | 18 | 17 | 19 | 17 | 71 | 21.78% |
| Bahrain | 28 | 22 | 28 | 26 | 104 | 31.90% |
| Kuwait | 8 | 6 | 9 | 8 | 31 | 9.51% |
| Total | 88 | 74 | 84 | 80 | 326 | 100.00% |

*4.2. Data Analysis*

OLS regressions of profitability measures, including Return on Assets and Return on Equity on governance measures, institutional variables, and control variables, were conducted during the pre-crisis and during crisis periods. The models are specified below,

$$ROA = \alpha_0 + \quad \beta_1 CD_i + \beta_2 BS_i + \beta_3 BI_i + \beta_4 OC_i + \beta_5 IO_i + \beta_6 FO_i + \beta_7 CO_i + \beta_8 PC_i + \beta_9 MC_i \\ + \beta_{10} BSI + \beta_{11} LR_i + \beta_{12} GD_i + \varepsilon \tag{1}$$

$$ROE = \alpha_0 + \quad \beta_1 CD_i + \beta_2 BS_i + \beta_3 BI_i + \beta_4 OC_i + \beta_5 IO_i + \beta_6 FO_i + \beta_7 CO_i + \beta_8 PC_i + \beta_9 MC_i + \\ \beta_{10} BSI + \beta_{11} LR_i + \beta_{12} GD_i + \varepsilon \tag{2}$$

A Generalized Method of Moments model, and Two-Stage Least Squares model, acted as a robustness check on the models specified in Equations (1) and (2). Where, *CD* = CEO Duality, it is a dichotomous variable, with value 1 if the CEO is the Chair of the Board, *BS* = Size of the board, measured by the total number of board members, *BI* = Percentage of independent members on the board of directors, *OC* = Ownership Concentration, measured by the percentage of shares owned by the CEO ad executives, *IO* = Institutional Ownership Concentration, measured by the percentage of shares owned by institutional investors, *FO* = Foreign Ownership Concentration, measured by the percentage of shares owned by foreign investors, *CO* = Performance-based Compensation, with a value of 1 if the bank implements performance-based compensation for executives, and 0 if there is no performance-based compensation, *PC* = Ownership Political Concentration, or the percentage of shares owned by politicians, *MC* = Managerial Political Connection, which equals 1 for involvement by politicians with managers, or the board of directors, *BSI* = Bank Size, measured by Total Assets, *LR* = Liquidity Ratio, measured by Total value of loans/Total value of deposits, *GD* = Gross Domestic Product.

**5. Results**

*5.1. Descriptive Statistics*

Table 3 shows the descriptive statistics of the dependent and independent variables listed in Section 4. The average return of banks, in terms of *ROA* and *ROE*, decreased in 2020, although some recovery was observed in 2021. Although corporate governance variables, and bank size, were largely stable, both the liquidity ratio and GDP decreased in 2020, with some reversal in 2021.

**Table 3.** Descriptive Statistics.

| Variable | 2018 | | 2019 | | 2020 | | 2021 | |
|---|---|---|---|---|---|---|---|---|
| | Average | SD | Average | SD | Average | SD | Average | SD |
| Return on assets (*ROA*) | 0.2523 | 0.7461 | 0.2433 | 1.4141 | −0.2401 | 1.3313 | 0.1123 | 1.1041 |
| Return on equity (*ROE*) | 3.4151 | 3.0193 | 3.1091 | 1.5631 | 1.8316 | 1.0196 | 2.0013 | 1.2111 |
| Duality | 0.6341 | 0.1271 | 0.6782 | 0.1423 | 0.6725 | 0.1183 | 0.6591 | 0.1129 |
| Size of board | 9.4228 | 1.3575 | 9.2384 | 1.5031 | 9.4745 | 1.4838 | 9.3353 | 1.002 |
| Independent members | 0.4245 | 0.0938 | 0.3984 | 0.1093 | 0.4021 | 0.0922 | 0.4221 | 0.1793 |
| Ownership concentration | 0.4524 | 0.2018 | 0.5534 | 0.1817 | 0.5048 | 0.1197 | 0.4907 | 0.2091 |
| Institutional ownership concentration | 0.1039 | 0.0736 | 0.1118 | 0.0664 | 0.1020 | 0.0969 | 0.1109 | 0.1034 |
| Foreign ownership concentration | 0.0915 | 0.0392 | 0.0892 | 0.0471 | 0.0884 | 0.0731 | 0.0736 | 0.0116 |
| Compensation based performance | 0.4164 | 0.0475 | 0.3651 | 0.0661 | 0.4094 | 0.0558 | 0.3852 | 0.0452 |
| Ownership political concentration | 0.3541 | 0.1203 | 0.3452 | 0.1038 | 0.3526 | 0.1120 | 0.3603 | 0.1208 |
| Managerial political connection | 0.31191 | 0.1142 | 0.31031 | 0.1156 | 0.3094 | 0.1294 | 0.3192 | 0.1394 |
| Size of bank | 7.2481 | 0.4663 | 7.0494 | 0.4021 | 7.1028 | 0.3985 | 7.15965 | 0.4013 |
| Liquidity ratio | 74.9481 | 7.3612 | 70.3729 | 10.3944 | 67.4831 | 11.4094 | 68.4213 | 9.4049 |
| Gross Domestic Product | 1.7936 | 0.6263 | 0.83426 | 1.6628 | −5.0666 | 2.1986 | 2.3232 | 0.6421 |

*5.2. Descriptive Statistics: t-Tests*

Table 4 elaborates upon the descriptive statistics, first presented in Table 3. *t*-tests measuring bank performance before the crisis and during the crisis. Both *ROA* and *ROE* were significantly lower during the crisis period than in the pre-crisis period. For *ROA*, mean differences of 0.3117, $t = 5.5038$, $p < 0.001$ were obtained, while for *ROE*, the mean difference of 7.70463, $t = 7.0462$, $p < 0.001$ was observed. Likewise, the institutional variable of the liquidity ratio was lower during the crisis. The mean difference was 4.7083, $t = -4.1682$, $p < 0.001$. The control variable of GDP also showed a significant decline during the pandemic. All other corporate governance variables and institutional variables had insignificant mean differences before and during the crisis.

**Table 4.** *t*-Test Results.

| Ratio | Period | Mean | N | Mean Difference | *t* | *t*-Test Sig (2 Tailed) |
|---|---|---|---|---|---|---|
| Return on Assets | Before crisis | 0.2478 | 162 | 0.3117 | 5.5038 | 0.0000 |
| | During crisis | −0.0639 | 164 | 0.3117 | 5.9243 | |
| Return on Equity | Before crisis | 3.2621 | 162 | 1.34565 | 7.0462 | 0.0000 |
| | During crisis | 1.91645 | 164 | 1.34565 | 7.9433 | |
| Duality | Before crisis | 0.65615 | 162 | −0.00965 | −2.5531 | 0.1127 |
| | During crisis | 0.6658 | 164 | −0.00965 | −2.5521 | |
| Size of board | Before crisis | 9.3306 | 162 | −0.0743 | −1.22321 | 0.3234 |
| | During crisis | 9.4049 | 164 | −0.0743 | −1.55795 | |
| Independent members | Before crisis | 0.41145 | 162 | −0.00065 | −4.42453 | 0.4521 |
| | During crisis | 0.4121 | 164 | −0.00065 | −4.43031 | |
| Ownership concentration | Before crisis | 0.5029 | 162 | 0.00515 | −0.26421 | 0.7944 |
| | During crisis | 0.49775 | 164 | 0.00515 | −0.25328 | |
| Institutional ownership concentration | Before crisis | 0.10785 | 162 | 0.0014 | −4.40558 | 0.3311 |
| | During crisis | 0.10645 | 164 | 0.0014 | −4.18522 | |
| Foreign ownership concentration | Before crisis | 0.09035 | 162 | 0.00935 | −7.04236 | 0.2421 |
| | During crisis | 0.08102 | 164 | 0.00935 | −8.53423 | |
| Compensation based performance | Before crisis | 0.39075 | 162 | 0.03802 | −2.23371 | 0.3741 |
| | During crisis | 0.35273 | 164 | 0.03802 | −2.14423 | |

**Table 4.** *Cont.*

| Ratio | Period | Mean | N | Mean Difference | *t* | *t*-Test Sig (2 Tailed) |
|---|---|---|---|---|---|---|
| Ownership political concentration | Before crisis | 0.34965 | 162 | −0.0068 | −9.22441 | 0.2566 |
| | During crisis | 0.35645 | 164 | −0.0068 | −9.57529 | |
| Managerial political connection | Before crisis | 0.31141 | 162 | −0.00319 | −4.41323 | 0.3421 |
| | During crisis | 0.31463 | 164 | −0.00319 | −4.41041 | |
| Size of bank | Before crisis | 7.14875 | 162 | 0.017525 | −0.26441 | 0.7945 |
| | During crisis | 7.13122 | 164 | 0.017525 | −0.25865 | |
| Liquidity ratio | Before crisis | 72.6605 | 162 | 4.7083 | −4.4034 | 0.0000 |
| | During crisis | 67.9522 | 164 | 4.7083 | −4.1682 | |
| Gross Domestic Product | Before crisis | 1.31393 | 162 | 2.68563 | −7.0464 | 0.0000 |
| | During crisis | −1.37173 | 164 | 2.68563 | −8.5335 | |

### 5.3. Results of Hypotheses Testing: The Fixed Effects Model

Table 5 shows the results of fixed-effects models used to test hypotheses. Hypothesis 1 was supported as the presence of independent members on the board of directors significantly increased the return on assets and the return on equity, both before the COVID-19 crisis and during the COVID-19 crisis. Pre-crisis coefficients were 0.12, $p < 0.05$ for *ROA*, and 0.13, $p < 0.01$, for *ROE*. During the crisis, significant coefficients of 0.24 for *ROA*, $p < 0.01$, and 0.24, $p < 0.05$, for *ROE,* were found. Hypothesis 2 was partly supported, as ownership political concentration resulted in increases in *ROA*, and *ROE* in both the pre-crisis, and crisis periods. Pre-crisis coefficients were 0.12, $p < 0.05$ for *ROA*, and 0.12, $p < 0.05$, for *ROE*. During the crisis, significant coefficients of 0.33 for *ROA*, $p < 0.01$, and 0.24, $p < 0.01$, for *ROE* were found. Foreign ownership concentration and institutional ownership concentration did not significantly influence *ROA* and *ROE*. Hypothesis 3 was rejected as managerial political connection failed to significantly influence *ROA* and *ROE* in both periods. Hypothesis 4 was supported during the crisis period, as bank size significantly influenced *ROA* (Coefficient = 0.05, $p < 0.01$) and *ROE* (Coefficient = 0.05, $p < 0.01$). Hypothesis 5 was fully supported for both periods, as the liquidity ratio significantly increased return on assets and on equity in both periods. Pre-crisis coefficients were 0.29, $p < 0.01$ for *ROA*, and 0.28, $p < 0.05$, for *ROE*. During the crisis, significant coefficients of 0.36 for *ROA*, $p < 0.05$, and 0.34, $p < 0.01$, for *ROE,* were found.

**Table 5.** OLS Regressions of *ROA* and *ROE* on Corporate Governance Variables, and Institutional Variables Before the COVID-19 Crisis, and During the COVID-19 Crisis.

| Variables | Before Crisis | | During Crisis | |
|---|---|---|---|---|
| | **Return on Assets** | **Return on Equity** | **Return on Assets** | **Return on Equity** |
| | Coefficient | Coefficient | Coefficient | Coefficient |
| (Constant) | 1.64511 *** | 1.75315 *** | 1.75205 *** | 1.76804 *** |
| Duality | −0.05887 | −0.08342 | −0.04673 | −0.05730 |
| Size of board | −0.06134 | −0.07926 | −0.05742 | −0.04516 |
| Independent members | 0.12241 * | 0.13013 ** | 0.24535 ** | 0.24827 * |
| Ownership concentration | 0.03754 | 0.03572 | 0.04882 | 0.05312 |
| Institutional ownership concentration | 0.12528 | 0.13029 | 0.10048 | 0.10193 |
| Foreign ownership concentration | 0.04521 | 0.03481 | 0.05471 | 0.04842 |
| Compensation based performance | 0.21446 | 0.22471 | 0.35627 | 0.36235 |
| Ownership political concentration | 0.12725 * | 0.11057 * | 0.33742 ** | 0.24525 ** |
| Managerial political connection | 0.04263 | 0.03982 | 0.05682 | 0.09227 |
| Size of bank | 0.0572 | 0.0420 | 0.05746 ** | 0.05562 ** |
| Liquidity ratio | 0.29474 ** | 0.28456 * | 0.36252 * | 0.34480 ** |

**Table 5.** *Cont.*

| Variables | Before Crisis | | During Crisis | |
|---|---|---|---|---|
| | Return on Assets | Return on Equity | Return on Assets | Return on Equity |
| | Coefficient | Coefficient | Coefficient | Coefficient |
| Gross Domestic Product | 0.16347 | 0.17839 | 0.20333 | 0.22862 |
| $R^2$ | 0.54292 | 0.63325 | 0.67363 | 0.53325 |
| Adj. $R^2$ | 0.49662 | 0.45256 | 0.53352 | 0.54928 |
| F-stat | 6.84256 | 6.84799 | 7.23315 | 6.84442 |

Note: * Significant at 10%; ** significant at 5%; *** significant at 1%.

### 5.4. Robustness Checks

A Generalized Method of Moments (GMM) estimator model and a Two-Stage Least Squares Model were employed to reveal problems of endogenous independent variables, detect and solve for heteroscedasticity, and uncover any unobserved independent variables. By introducing lags of the independent variables as instruments, we were able to filter out the endogeneity of independent variables. High $R^2 > 0.50$ indicated that the models were well-specified, with no additional independent variables. Heteroscedasticity was undetected. The empirical findings of GMM estimators in Table 6 show the same results of fixed-effect models shown in Table 6. Therefore, the standard errors detected in the regression models are unbiased.

**Table 6.** GMM Regressions of *ROA* and *ROE* on Corporate Governance Variables and Institutional Variables Before the COVID-19 Crisis and During the COVID-19 Crisis. Instrumental Variable = Lags of Corporate Governance Variables.

| Variables | Before Crisis | | During Crisis | |
|---|---|---|---|---|
| | Return on Assets | Return on Equity | Return on Assets | Return on Equity |
| | Coefficient | Coefficient | Coefficient | Coefficient |
| Duality | −0.05425 | −0.07837 | −0.06442 | −0.04532 |
| Size of board | −0.05242 | −0.06938 | −0.05562 | −0.03446 |
| Independent members | 0.13541 * | 0.12857 ** | 0.23225 ** | 0.31321 * |
| Ownership concentration | 0.0342i | 0.03442 | 0.04656 | 0.06212 |
| Institutional ownership concentration | 0.13511 | 0.12559 | 0.12094 | 0.12331 |
| Foreign ownership concentration | 0.03521 | 0.02955 | 0.05331 | 0.03721 |
| Compensation based performance | 0.19446 | 0.21848 | 0.28761 | 0.33119 |
| Ownership political concentration | 0.14323 * | 0.13958 * | 0.29481 ** | 0.20291 ** |
| Managerial political connection | 0.03256 | 0.034958 | 0.03582 | 0.04207 |
| Size of bank | 0.05332 | 0.03986 | 0.06091 ** | 0.05312 ** |
| Liquidity ratio | 0.23522 ** | 0.277262 * | 0.34551 * | 0.32934 ** |
| Gross Domestic Product | 0.14245 | 0.16474 | 0.20848 | 0.21602 |
| AR(1)-P | 0.0033 | 0.0552 | 0.0076 | 0.0214 |
| AR(2)-P | 0.3251 | 0.6094 | 0.0852 | 0.0528 |

Note: * $p < 0.05$, ** $p < 0.01$.

The Two-Stage Least Squares model in Table 7 indicates that the impact of independent variables on the performance of banks in GCC has the same impact as fixed-effect models. Therefore, both robustness checks conclude that the endogeneity bias of predictor variables is absent, as are missing variables that could bias the relation between corporate governance mechanisms and banks' financial profitability in GCC during a crisis.

**Table 7.** Two-Stage Least Squares Regressions of *ROA* and *ROE* on Corporate Governance Variables and Institutional Variables Before the COVID-19 Crisis and During the COVID-19 Crisis.

| Variables | Before Crisis | | During Crisis | |
|---|---|---|---|---|
| | Return on Assets | Return on Equity | Return on Assets | Return on Equity |
| | Coefficient | Coefficient | Coefficient | Coefficient |
| Duality | −0.04847 | −0.07645 | −0.04531 | −0.04726 |
| Size of board | −0.05938 | −0.06765 | −0.04612 | −0.03981 |
| Independent members | 0.09383 * | 0.12986 ** | 0.23410 ** | 0.24726 ** |
| Ownership concentration | 0.02955 | 0.03462 | 0.04674 | 0.05462 |
| Institutional ownership concentration | 0.13483 | 0.12054 | 0.12019 | 0.12209 |
| Foreign ownership concentration | 0.03902 | 0.03331 | 0.06095 | 0.05039 |
| Compensation based performance | 0.23919 | 0.21986 | 0.34339 | 0.34929 |
| Ownership political concentration | 0.13232 * | 0.12098 * | 0.34421 ** | 0.23911 ** |
| Managerial political connection | 0.03976 | 0.03770 | 0.05161 | 0.05837 |
| Size of bank | 0.05210 | 0.04320 | 0.04872 ** | 0.04985 ** |
| Liquidity ratio | 0.29365 ** | 0.24521 * | 0.35773 * | 0.33019 ** |
| Gross Domestic Product | 0.15881 | 0.16736 | 0.20419 | 0.21983 |
| $R^2$ | 0.53201 | 0.52215 | 0.66746 | 0.540918 |
| Adj. $R^2$ | 0.48552 | 0.46736 | 0.54562 | 0.50071 |

Note: * Significant at 10%; ** significant at 5%.

The Mann–Whitney U test was employed to enhance the robustness of mean differences in the *t*-test (Table 4). The Mann–Whitney U test can be applied to a range of different data sets which means that there are no norms made about the distribution of the data. The result of the U test confirms the *t*-test results indicating that return on equity, return on assets, liquidity, and Gross Domestic Product were affected negatively during the COVID-19 pandemic period. The governance variables were not influenced by the development of the pandemic in GCC countries (see Table 8).

**Table 8.** Mann–Whitney U Test Results.

| Ratio | Period | Mean | N | Mean Difference | Z-Score | *p*-Value |
|---|---|---|---|---|---|---|
| Return on Assets | Before crisis | 0.2478 | 162 | 0.3117 | 3.4159 | 0.0002 |
| | During crisis | −0.0639 | 164 | | | |
| Return on Equity | Before crisis | 3.2621 | 162 | 1.3456 | 3.7227 | 0.0000 |
| | During crisis | 1.91645 | 164 | | | |
| Duality | Before crisis | 0.65615 | 162 | −0.0096 | 2.3704 | 0.6011 |
| | During crisis | 0.6658 | 164 | | | |
| Size of board | Before crisis | 9.3306 | 162 | −0.0743 | 3.4720 | 0.5918 |
| | During crisis | 9.4049 | 164 | | | |
| Independent members | Before crisis | 0.41145 | 162 | −0.0006 | 2.3122 | 0.5220 |
| | During crisis | 0.4121 | 164 | | | |
| Ownership concentration | Before crisis | 0.5029 | 162 | 0.0051 | 2.1183 | 0.6392 |
| | During crisis | 0.49775 | 164 | | | |
| Institutional ownership concentration | Before crisis | 0.10785 | 162 | 0.0014 | 2.5412 | 0.4354 |
| | During crisis | 0.10645 | 164 | | | |
| Foreign ownership concentration | Before crisis | 0.09035 | 162 | 0.0093 | 2.4325 | 0.5235 |
| | During crisis | 0.08102 | 164 | | | |
| Compensation based performance | Before crisis | 0.39075 | 162 | 0.0380 | 2.1847 | 0.5464 |
| | During crisis | 0.35273 | 164 | | | |
| Ownership political concentration | Before crisis | 0.34965 | 162 | −0.0068 | 2.5492 | 0.6781 |
| | During crisis | 0.35645 | 164 | | | |

**Table 8.** *Cont.*

| Ratio | Period | Mean | N | Mean Difference | Z-Score | *p*-Value |
|---|---|---|---|---|---|---|
| Managerial political connection | Before crisis | 0.31141 | 162 | −0.0031 | 3.8371 | 0.7455 |
| | During crisis | 0.31463 | 164 | | | |
| Size of bank | Before crisis | 7.14875 | 162 | 0.0175 | 2.2311 | 0.5317 |
| | During crisis | 7.13122 | 164 | | | |
| Liquidity ratio | Before crisis | 72.6605 | 162 | 4.7083 | 3.7170 | 0.0000 |
| | During crisis | 67.9522 | 164 | | | |
| Gross Domestic Product | Before crisis | 1.31393 | 162 | 2.6856 | 3.1592 | 0.0000 |
| | During crisis | −1.37173 | 164 | | | |

Finally, to disentangle the effects of corporate governance and institutional characteristics on banks' performance before and during the COVID-19 crisis, we include a dummy variable (COVID) and then we test the following regressions based on the whole sample period (see Table 9):

Regression (1):

$$OE = \alpha_0 + \begin{aligned} &\beta_1 CD_i + \beta_2 BS_i + \beta_3 BI_i + \beta_4 OC_i + \beta_5 IO_i + \beta_6 FO_i + \beta_7 CO_i + \beta_8 PC_i + \beta_9 MC_i \\ &+ \beta_{10} BSI + \beta_{11} LR_i + \beta_{12} GD_i + \beta_{13} CD_i * COVID + \beta_{14} BS_i * COVID + \beta_{15} BI_i \\ &* COVID + \beta_{16} OC_i * COVID + \beta_{17} IO_i * COVID + \beta_{18} FO_i * COVID + \beta_{19} CO_i \\ &* COVID + \beta_{20} PC_i * COVID + \beta_{21} MC_i * COVID + \beta_{22} BSI * COVID + \beta_{23} LR_i \\ &* COVID + \beta_{24} GD_i * COVID + \varepsilon \end{aligned}$$

Regression (2):

$$ROE = \alpha_0 + \begin{aligned} &\beta_1 CD_i + \beta_2 BS_i + \beta_3 BI_i + \beta_4 OC_i + \beta_5 IO_i + \beta_6 FO_i + \beta_7 CO_i + \beta_8 PC_i + \beta_9 MC_i \\ &+ \beta_{10} BSI + \beta_{11} LR_i + \beta_{12} GD_i + \beta_{13} CD_i * COVID + \beta_{14} BS_i * COVID + \beta_{15} BI_i \\ &* COVID + \beta_{16} OC_i * COVID + \beta_{17} IO_i * COVID + \beta_{18} FO_i * COVID + \beta_{19} CO_i \\ &* COVID + \beta_{20} PC_i * COVID + \beta_{21} MC_i * COVID + \beta_{22} BSI * COVID + \beta_{23} LR_i \\ &* COVID + \beta_{24} GD_i * COVID + \varepsilon \end{aligned}$$

**Table 9.** OLS Regressions of *ROA* and *ROE* on Corporate Governance Variables and Institutional Variables Before the COVID-19 Crisis and During the COVID-19 Crisis including COVID as dummy variable.

| Variables | Return on Assets | Return on Equity |
|---|---|---|
| | Coefficient | Coefficient |
| (Constant) | 1.72974 *** | 1.75852 *** |
| Duality | −0.05602 | −0.08763 |
| Duality × COVID | −0.04310 | −0.06372 |
| Size of board | −0.05947 | −0.07643 |
| Size of board × COVID | −0.04869 | −0.05167 |
| Independent members | 0.10194 * | 0.140941 *** |
| Independent members × COVID | 0.19001 *** | 0.26847 *** |
| Ownership concentration | 0.02918 | 0.041038 |
| Ownership concentration × COVID | 0.03736 | 0.04928 |
| Institutional ownership concentration | 0.11093 | 0.12117 |
| Institutional ownership concentration × COVID | 0.09378 | 0.12152 |
| Foreign ownership concentration | 0.05094 | 0.03646 |
| Foreign ownership concentration × COVID | 0.06118 | 0.03916 |
| Compensation based performance | 0.19009 | 0.21038 |
| Compensation based performance × COVID | 0.40194 | 0.36793 |
| Ownership political concentration | 0.11768 ** | 0.12002 * |
| Ownership political concentration × COVID | 0.30184 * | 0.22074 ** |
| Managerial political connection | 0.02087 | 0.04013 |

**Table 9.** *Cont.*

| Variables | Return on Assets | Return on Equity |
|---|---|---|
| | Coefficient | Coefficient |
| Managerial political connection × COVID | 0.05718 | 0.11709 |
| Size of bank | 0.06323 | 0.04517 |
| Size of bank × COVID | 0.06090 ** | 0.04986 ** |
| Liquidity ratio | 0.30015 * | 0.29018 * |
| Liquidity ratio × COVID | 0.29508 * | 0.28775 ** |
| Gross Domestic Product | 0.15482 | 0.18421 |
| Gross Domestic Product × COVID | 0.19484 | 0.19902 |
| $R^2$ | 0.63140 | 0.68571 |
| Adj. $R^2$ | 0.52012 | 0.56831 |
| F-stat | 7.14819 | 7.45784 |

Note: * $p < 0.05$, ** $p < 0.01$, *** $p < 0.001$.

The results of the Difference-in-Difference analysis reveal almost the same outputs as presented in Tables 5 and 6. The presence of independent members on the board of directors and the ownership concentration of politicians supported banks in sustaining their returns during the crisis. The regression analysis also provides evidence that banks of larger size and higher liquidity succeeded in facing the economic drop during the pandemic.

## 6. Conclusions and Discussion

### 6.1. Summary of Findings

The findings of this study may be summarized as follows.

1.  For GCC banks, independent directors on the board increased the return on assets and the return on equity in the pre-crisis and crisis periods.
2.  For GCC banks, ownership's political concentration increased the return on assets and the return on equity in the pre-crisis and crisis periods.
3.  For GCC banks, managerial political connections had no effect on the return on assets and the return on equity in the pre-crisis and crisis periods.
4.  For GCC banks, bank size increased the return on assets and the return on equity during the crisis period.
5.  For GCC banks, the liquidity ratio increased the return on assets and the return on equity in the pre-crisis and crisis periods.

### 6.2. Theoretical Implications

This paper has expanded our knowledge of the influence of corporate governance variables on profitability by conducting an empirical examination during a financial crisis, i.e., the COVID-19 lockdown of 2020–2021. Although up to eleven corporate governance variables and institutional variables have predicted profitability in the literature, during the crisis, just four variables acted as predictors. They included the presence of independent directors, ownership's political concentration, bank size, and liquidity ratio. Each of these findings will be discussed in depth in the following sub-sections.

#### 6.2.1. Independent Directors

The importance of independent directors in the GCC is similar to the finding by El-Chaarani et al. (2022) in the MENA region of independent directors significantly influencing both return on assets and return on equity during the COVID-19 lockdown. Independent directors may have played contrasting roles in both studies. In the MENA region, the lower income of bank depositors and borrowers, along with the findings of Mdaghri (2022) and Sahyouni and Wang (2019), suggest that there is a large volume of nonperforming loans. Independent directors may be engaged in monitoring management's ability to reduce nonperforming loans. In the GCC region, higher incomes suggest that the emphasis may be on maintaining the accounts of large depositors. Independent directors monitor the ability

of managers to offer higher interest rates for large deposits, waive fees and charges on new accounts with large balances, and create new deposit instruments, such as certificates of deposits, that offer competitive interest rates.

### 6.2.2. Ownership Political Concentration

We extend Boussada and Hakimi's (2021) finding of the significant positive impact of the presence of large shareholders on return on assets and return on equity in the MENA region to the GCC countries. We specify that it is the ownership's *political* concentration that increases profitability during crises, as institutional political concentration and foreign ownership concentration yielded no impact on profitability. The presence of politicians on bank boards, be they retired military, or former ministers, is beneficial, as they have direct access to members of government. During the pandemic, certain businesses received large government bailouts, including banks. Such politician-directors could have solicited bailout funds for their own businesses. Such funds were converted into new loans, which increased bank net interest income and bank profitability. Further, these individuals helped to maintain relationships with existing large depositors, who may have been family members or close friends.

### 6.2.3. Bank Size

The significant effect of bank size on profitability was confined to the crisis period. During the crisis, large banks continued to offer loans due to their large reserves. Although loan interest income was sufficient in the pre-crisis period, it was insufficient to sustain bank profits during the crises. Size helped large banks to locate alternate revenue streams, such as loan commitments, derivatives, and letters of credit.

### 6.2.4. Liquidity Creation

Liquidity creation has been recognized as a core function of banks in that banks take funds from depositors. Liquidity creation was a more important predictor of return on assets before the crisis, although it became a more important predictor of return on equity during the crisis. Return on equity is returned to the shareholders. As creating loans was more challenging during the crisis, banks that created such loans may have been viewed by shareholders as creating wealth. The equity of such investors may have risen in value as their confidence in bank loan creation increased.

### *6.3. Other Theoretical Implications*

To our knowledge, there is no study that showed the interaction between political connections and corporate governance mechanisms and its impact on the financial performance of the GCC banking sector. Thus, this research fills this gap and extends the research work on corporate governance mechanisms in banking firms. Furthermore, the previous studies were performed based on small samples of selected banks in the GCC region without considering the crisis periods. Therefore, this study contributed to the corporate governance field by using a larger sample studied during the crisis period. Finally, this study provides a partial confirmation of several theories such as agency and entrenchment theory. Agency costs occur with managerial political connections. However, ownership's political concentration, and independent directors, overcome these agency costs, as both of these variables significantly increase profits. Managerial entrenchment may result in revenue-reducing strategies. However, independent directors' monitoring of management's ability to make rational decisions during crises may have prevented such losses.

The finding that, during the COVID-19 crisis, the GCC did not benefit from a compensation committee or CEO duality, as observed during the 2007–2009 financial crisis (Ayadi et al. 2019), is significant. These corporate governance mechanisms contributed to the hiring of resourceful CEOs to cope with the 2007–2009 financial crisis. In the GCC

countries during the COVID-19 lockdown, the political connections of large shareholders in obtaining deposits may have been the more effective coping strategy.

### 6.4. Practical and Policy Implications

This paper has several implications for bankers, governments, and financial regulators.

Bankers know that preparation for the next financial crisis is essential. GCC banks must place independent directors on the board and encourage large shareholders to use their political connections to obtain access to government resources. As bank size is important during crises, small banks may form consortiums with mid-size banks so that they will have the funds to continue to make loans, even in times of economic instability.

Above all, banks must strive to maintain liquidity, or sufficiently large cash reserves, to be able to make loans during regular periods and crises. It is loan creation that provides the interest income that permits banks to perform their essential role of financial intermediation. As mentioned, smaller banks must seek partnerships with larger entities that will enable them to access funds. Government programs targeted at small banks may provide a steady stream of funds. Banks are encouraged to apply for such government funding, even during prosperous periods, so that they become established borrowers. Large banks may have reserves or political connections to obtain these funds.

### 6.5. Limitations and Research Plans

This study has several limitations that could be considered in future research papers. First, this paper is based on a small sample and considers a very short period. Only four years and 88 banks were considered in this study; thus, future research papers must be performed by considering larger samples and periods.

Second, in regression models, this paper does not consider the impact of corporate governance and political connections within each country in the GCC. Therefore, future work can be performed by studying the impact of corporate governance and political connections on the financial profitability within each GCC member country.

The employed dependent variables could be extended to include other variables related to market risks and market performance. Finally, future research papers could include other corporate governance variables, such as legal protection and takeover strategy.

**Author Contributions:** Conceptualization: H.E.-C., D.K. and M.S.-A., Methodology: H.E.-C., D.K. and M.S.-A., Validation: H.E.-C. and R.A., Formal Analysis: H.E.-C., Investigation: H.E.-C., Resources: D.K. and M.S.-A., Data Curation: H.E.-C., Writing: Original Draft Preparation: R.A., Writing: Review and Editing, H.E.-C. and R.A., Visualization: H.E.-C., Supervision: H.E.-C., Project Administration: R.A. All authors have read and agreed to the published version of the manuscript.

**Funding:** This research received no external funding.

**Data Availability Statement:** Data is available from the first author upon request.

**Conflicts of Interest:** The authors declare no conflict of interest.

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
