# Peer review of "Corporate Governance Effects on Bank Profits in Gulf Cooperation Council Countries during the Pandemic"

_ijfs, doi:10.3390/ijfs11010036_

Round 1
Reviewer 1 Report
Referee report on
IJFS-2181604
The Influence of Corporate Governance and Institutional Variables on Bank Profitability in GCC Countries: An Assessment of The Covid-19 Lockdown Period
Submitted to the International Journal of Financial Studies
This paper analyses the effect of corporate governance characteristics of ownership political concentration, managerial political connection, independent directors and institutional variables on bank performance in GCC countries before and during the COVID-19 pandemic. The author(s) conclude that the bank performance is positively affected by ownership political concentration, independent directors, bank size and liquidity. I think that the paper is interesting and focuses on a topic relevant in the field of banking and finance. The investigation of the role of COVID-19 pandemic on the performance of banks considering the effect of corporate governance and institutional variables is also interesting. However, the paper has some issues for which I provide several suggestions to the author(s) to improve their work.
Introduction / Contribution to literature
1. The motivation of the paper could be more compelling. The author(s) mainly put emphasis on the unique role of banking sector in the GCC countries and on the impact of political connections on bank performance. I think that to better establish its contribution, the paper would benefit from further explaining some points. Can author(s) justify in detail why there is a need for investigation of banking performance in GCC countries during the COVID-19 pandemic? Which are the new insights that this study brings to what we already know about the determinants of bank performance during times of market crises such as during the COVID-19 pandemic? Author(s) should explain better their motivation.
Review of Literature
2. A good variety of literature was used to justify the research objectives and to show the study’s context. An up to date account of literature was provided. However, the article may benefit from some debates on similar topics with respect to the determinants (including corporate governance mechanisms and institutional characteristics) of banking performance during periods of financial crises.
3. The author(s) fail to provide a detailed hypotheses development and do not fully reflect through which channels corporate governance mechanisms and institutional variables affect the banking performance. I think the authors should work more on this topic. For example, author(s) could focus more on the exact mechanisms that might be at stake: How and why do ownership political concentration, independent directors, bank size, and liquidity ratio affect the bank performance. It is very important that such a transmission mechanism is clearly spelled out and motivated and that theoretical justifications are provided for each hypothesis tested.
Methodology /Results
4. In table 3, except from using the parametric t-test for the comparison of key variables before and during the COVID-19 crisis, author(s) should enhance the robustness of their results by additionally using a non-parametric test (i.e. Mann-Whitney U test).
5. Minor: Please check again the performance ratios in the table 3. I think that the 2nd test of significance is referred to the Return on Equity ratio.
6. The results from the cross sectional analyses based only on separate regression models for the period before the COVID-19 pandemic and the period during the COVID-19 pandemic. This procedure fails to provide a more comprehensive analysis on whether and how the COVID-19 pandemic affect the association between corporate governance/institutional variables and bank performance. To disentangle the effects of corporate governance and institutional characteristics on bank performance before and during the COVID crisis, wouldn’t it be better to use a Difference-in-Difference analysis? I would like to see such an approach. By including a dummy variable that captures the COVID-19 pandemic and by applying Difference-in-Difference analysis for the whole sample period, important insights for can be derived. For example, in each of the tables 4 to 6, author(s) should add column(s) to present the results derived from the Difference-in-Difference analysis. The sign and the magnitude of the dummy variable assess the impact of the COVID-19 pandemic on bank performance, after controlling for corporate governance characteristics and institutional variables, whilst the Difference-in-Difference analysis can fully clarify whether and how the COVID-19 pandemic alter the effect of corporate governance or institutional characteristics on the bank performance.
I wish you best of luck in revising the paper.
Author Response
Please see the attached file. Reviewer 1 concerns are addressed in the Table. The document contains the changes to meet Reviewer 1's concerns in green.

Reviewer 2 Report
This paper requires copy editing and professional proofreading, throughout. Below are some comments to address:
The title should be improved.
1. The abstract should have one sentence per each: context and background, motivation, gaps of study, methods, results, conclusions, significance, and originality. The author (s) should provide a precise and focused abstract.
2. The introduction section has no fluency or clarity. The introduction section needs to be started with a broader area and issue or in a global context – then relate it with your topic, highlight the problems/issues in your area/context, and the proposed solution to address these issues.
3. Also, highlight the significance of the study in a precise and clear way in the introduction section. Authors should consult:
https://www.sciencedirect.com/science/article/pii/S0959652621023179
4. The paper incorporated major literature but does not sufficiently cover recent research in the area. Helpful in this regard would be to include relevant research recently covered in top journals of similar scope. Further, authors need to improve the theoretical discussions and need to highlight how this research is contributing to the literature. Figure 1 is not clear, it should be revised. Author(s) should consult:
https://www.tandfonline.com/doi/full/10.1080/13504509.2022.2134230
5. The methodology needs to be improved. Why the sample is covering from 2018 to 2021? It is confusing, as the authors mentioned the sample is from 2020 to 2021.
6. Please provide the findings/results of the study and then discuss their theoretical, practical, and policy implications. Please write the implication under the discussion section.
7. The conclusion should be precise and clear to summarize the study. It is suggested to write the conclusion in a precise and clear way.
8. The reviewer found that the authors have cited less recently published papers in this article, and the majority are very old. As a suggestion, the author must cite new articles (latest literature) to make a holistic discussion and sturdy paper with high readability.
I hope that the comments provided and papers suggested can help in this regard.
Author Response
Reviewer 2's concerns are listed in the table in the attached file. In the document, changes have been made in yellow.

Round 2
Reviewer 1 Report
Author(s) adequately adress several of the concerns raised. However, the difference-in-difference approach should be corrected. In the DID approach, the dummy "COVID" should also be added separately (without an interaction term) in order to capture the net effect of COVID-19 pandemic on the banking performance. Thus, author(s) should also include an extra variable “COVID” into the regression analyses of DID approach.
Author Response
We are unable to add variables to the regression, as the reliability of the regression will decrease.

Reviewer 2 Report
The revised version of the paper is not able to be processed for further actions, it still requires major revisions. In the revised version, the authors have not addressed any of my comments. They have only increased literature but did not improved it all. Only abstract and introduction are more than 8 pages. I suggest the authors to carefully read my previous comments and revise the paper accordingly.
Author Response
We have made the requested changes in yellow.
